# The Impact of Total Knee Replacement with a Customized Cruciate-Retaining Implant Design on Patient-Reported and Functional Outcomes

**DOI:** 10.3390/jpm12020194

**Published:** 2022-01-31

**Authors:** Andre F. Steinert, Lennart Schröder, Lukas Sefrin, Björn Janßen, Jörg Arnholdt, Maximilian Rudert

**Affiliations:** 1Department of Orthopaedic Surgery, König-Ludwig-Haus, Julius-Maximilians-University Würzburg, Brettreichstraße 11, D-97074 Würzburg, Germany; andre.steinert@campus-nes.de (A.F.S.); Lennart.Schroeder@med.uni-muenchen.de (L.S.); lsef@gmail.de (L.S.); bjoern_janssen@web.de (B.J.); Joerg.Arnholdt@med.uni-muenchen.de (J.A.); 2Rhön Klinikum, Campus Bad Neustadt, EndoRhön Center for Joint Replacement, Teaching Hospital of the Phillipps University Marburg, Von Guttenberg Str. 11, D-97616 Bad Neustadt, Germany; 3Department of Orthopaedics and Trauma Surgery, Musculoskeletal University Center Munich (MUM), University Hospital, Ludwigs-Maximilians-University Munich, Marchionistr. 15, D-81377 Munich, Germany

**Keywords:** patient-specific, custom-made implant, total knee arthroplasty, TKA, knee replacement, tricompartmental knee osteoarthritis, iTotal

## Abstract

Purpose: To treat patients with tricompartimental knee osteoarthritis (OA), a customized cruciate-retaining total knee arthroplasty (CCR-TKA) system can be used, including both individualized instrumentation and implants. The objective of this monocentric cohort study was to analyze patient-reported and functional outcomes in a series of patients implanted with the second generation of this customized implant. Methods: At our arthroplasty center, we prospectively recruited a cohort of patients with tricompartmental gonarthrosis to be treated with total knee replacement (TKA) using a customized cruciate-retaining (CCR) implant design. Inclusion criteria for patients comprised the presence of intact posterior cruciate and collateral ligaments and a knee deformity that was restricted to <15° varus, valgus, or flexion contracture. Patients were assessed for their range of motion (ROM), Knee Society Score (KSS), Western Ontario and McMaster University osteoarthritis index (WOMAC), and short form (SF)-12 physical and mental scores, preoperatively, at 3 and 6 months, as well as at 1, 2, 3, and 5 years of follow-up (FU) postoperatively. Results: The average age of the patient population was 64 years (range: 40–81), the average BMI was 31 (range: 23–42), and in total, 28 female and 45 male patients were included. Implant survivorship was 97.5% (one septic loosening) at an average follow-up of 2.5 years. The KSS knee and function scores improved significantly (*p* < 0.001) from, respectively, 41 and 53 at the pre-operative visit, to 92 and 86, respectively, at the 5-year post-operative time point. The SF-12 Physical and Mental scores significantly (*p* < 0.001) improved from the pre-operative values of 28 and 50, to 50 and 53 at the 5-year FU, respectively. Patients experienced significant improvements in their overall knee range of motion, from 106° at the preoperative visit to 122°, on average, 5 years postoperatively. The total WOMAC score significantly (*p* < 0.001) improved from 49.1 preoperatively to 11.4 postoperatively at 5-year FU. Conclusions: Although there was no comparison to other implants within this study, patients reported high overall satisfaction and improvement in functional outcomes within the first year from surgery, which continued over the following years. These mid-term results are excellent compared with those reported in the current literature. Comparative long-term studies with this device are needed. Level of evidence 3b (individual case–control study).

## 1. Introduction

Advanced knee osteoarthritis (OA) is a disabling disease frequently requiring knee replacement surgery. Despite overall improvements over the past few decades in total knee arthroplasty (TKA), surgical procedures, and implant design, recent studies have shown that approximately 19% of patients treated with TKA continue to experience discomfort in their treated joint [1,2,3]. Inappropriate size, fit, and positioning of the implant components including rotational and coronal alignment have emerged among the key factors that lead to a higher risk of implant failure, poor outcomes, and high revision rates over time [4].

To overcome these limitations, several novel surgical techniques in TKA surgery have been explored in recent years [5]. Patient-specific knee prostheses have been introduced to provide an ideal coverage of the bony surfaces of the tibia and the femur and are shaped to address the patient-specific J-curve anatomy of the bones [6,7]. The second generation (G2) of a patient-specific cruciate-retaining TKA system iTotal™ (CCR-TKA) comprises custom-made implants as well as instrumentation and represents a new approach for the treatment of patients with tricompartmental knee OA. Based on computed tomography (CT) scans of the affected limb and computer-aided design and manufacturing (CAD/CAM) protocols, this system aims to achieve an optimal fit of implant components and instruments [8,9]. Using this implant technology, encouraging initial clinical and radiographic results have been reported for unicompartmental (UKA) [10,11] and bicompartmental (BKA) knee arthroplasty [12,13,14].

Therefore, the aim of this prospective longitudinal clinical study was to analyze the clinical outcome of the treatment of tricompartmental knee OA with CCR-TKA with a follow-up of up to 5 years.

## 2. Methods

### 2.1. Patients

In this single-center study at a German university arthroplasty center, a cohort of 73 patients was recruited prospectively from November 2012 until January 2017 to undergo TKA with the iTotal^®^ CR G2 knee replacement (Conformis Inc., Billerica, MA, USA). Patients were diagnosed with end-stage tricompartmental osteoarthritis, and individuals with compromised posterior cruciate or collateral ligaments or having a varus/valgus deformity or fixed extensor lag >15° were excluded. Other exclusion criteria were: active local or systemic infection, immunodeficiency, RA or other forms of inflammatory joint disease, prior arthroplasty of the affected knee, and prior history of failed implant surgery of the joint to be treated, including high tibial osteotomy (HTO). An Ethics Committee approval was obtained from the Institutional Review Board (IRB) of the Julius-Maximilians University Medical Center (approval number 2016101401), and all patients signed an informed consent prior to participation. All surgeries were performed by two high-volume surgeons (first and senior author) using a standard medial parapatellar arthrotomy, under adherence to the standards of Good Clinical Practice (GCP). Patient enrollment and sampling were conducted to include subsequent cases willing to participate and to meet the inclusion criteria after a learning curve with this implant system of 6 months.

### 2.2. Custom Cruciate-Retaining TKA Implant and Planning

The CCR-TKA implant used has a CE marking and is approved by the United States Food and Drug Administration (FDA). A CT scan of the affected leg was conducted for every patient preoperatively by scanning the knee, the femoral head, and the talus center in accordance with a standard protocol (http://www.conformis.com/healthcare-professionals/imaging-professionals, accessed on 25 January 2022) as previously described [8,9]. The cemented, fixed-bearing, patient-specific implant was designed based on the patient’s bone geometry, defining shape and size of the metal implant components, as well as the disposable bone-cutting jigs [8]. A correction of the mechanical axes towards neutral, as well as a preservation of the joint line including the distal and the posterior femoral condylar offset and the tibial slope was implemented in the implant design process [7].

Representative images of the CCR-TKA knee implant are shown in Figure 1A–D. It comprises three components: a femoral shield, a tibial tray, and an optional patellar component, including a separate medial and lateral insert of the tibia with different heights that correspond to the condylar offsets of the femur. The femoral shield and tibial tray were manufactured using a cobalt–chromium–molybdenum alloy, the tibial inserts and the patellar component using ultra-high-molecular-weight polyethylene, while the disposable cutting jigs were made from nylon via 3D printing.

A typical 3D planning protocol (iView^®^) is shown in Figure 2, comprising representative images of the patient’s anatomic and implants’ features of the tibia (A) and femur (B), as well as the projected thicknesses of the respective tibial and femoral bone cuts for self-control (Figure 2A,B).

### 2.3. Surgical Technique

The detailed surgical procedure has been previously described [8]. In brief, following a medial parapatellar arthrotomy, the surgical procedure included 6 different steps, that were facilitated by the use of the provided patient-specific bone resection jigs and iView^®^ protocol, allowing confirmation of all performed bone cuts against the surgical plan for self-control. Specifically, the instrumentation kit comprised 6 different femoral (F1–6) and 5 separate tibial (T1–5) jigs for cutting and drilling. The surgical steps were distal femoral resection (required jigs: F1–3), proximal tibial resection (jig: T1), balancing of extension and flexion gap (jigs: T2, T3), femoral preparation (jigs: F4, F5), trialing (jigs: F6, T4), and final tibial preparation (jig: T5), before the final implantation of the components was performed. Notably, the surgeon had two options for the tibial cuts, with two different T1 instruments facilitating the tibial cut either with a patient-specific slope (shown in red) or with a fixed slope of 5° (shown in black), as shown in the tibial images of the iView^®^ (Figure 2A). Additionally, testing of the trial components could be performed with three individually designed T4 jigs with 1 mm incremental heights. Thereby, the CCR-TKA system allows for kinematic testing using anatomic trial components, where the most appropriate tibial insert heights for the medial (6 mm, 7 mm, or 8 mm) and lateral (A, B, C with custom thicknesses; see iView^®^ bottom row (Figure 2)) knee joint space may be identified before the final components are implanted. Patella resurfacing is performed if necessary, using standard oval dome patellae. Cementing, wound closure, and rehabilitation protocols were in accordance with standard TKA procedures.

### 2.4. Outcome Parameters

Several clinical outcome scores were assessed pre-operatively and compared to post-operative outcomes at 3 and 6 months, as well as at 1-, 2-, 3- and 5-year post-OP. The Knee Society Scoring System (KSS) by Insall et al. was assessed. It consists of 2 separate subscales: (1) a “Knee” score (100 points total) which considers pain (50 points), stability (25 points), and range of motion (25 points) with deductions for flexion contractures, extension lag, and malalignment, and (2) a “Function” score (100 points total) that utilizes walking distance (50 points) and stair climbing (50 points) with deduction for the use of a walking aid [15,16].

The Western Ontario and McMaster Universities Osteoarthritis Index (WOMAC) was used as a patient-reported outcome measure for knee osteoarthritis, that included the subscales “Pain” (5 items; 50 points), “Stiffness” (2 items; 20 points), and “Function” (17 items; 170 points), with a range from 0 (= no pain/stiffness/problems) to 10 (= extreme pain/stiffness/impossible to do) points for each item [17]. The relative WOMAC scores, i.e., for the total WOMAC score as well as for the subscales, were then calculated from the point values by multiplication × 100 and divided by the maximum score value [17,18].

We also assessed the Short Form (SF) 12 Health Survey, which is a 12-item, patient-reported survey of patient health, that evaluates eight dimensions of health status [19]. Scores in the range of 0–100 are given for each subscale, and higher scores represent better health. Norm-based scoring of each 0–100 scale is then carried out by standardization of each subscale relative to a Z-Score that is 50 on average in the population, with a standard deviation of 10 [19]. Finally, two aggregate summary measures can be derived, a physical (PCS) and a mental (MCS) health subscores, to determine the overall mental and physical well-being [19].

Any device-related adverse event that was observed or reported by the patient was evaluated, and implant survivorship was calculated for patients that had completed a follow-up of a minimum of 2 years post-OP or were revised prior to the 2-year time point.

Radiographic evaluations were also performed prior to surgery and one week post-surgery using a strict antero-posterior (AP) view, a lateral view (including a referencing sphere), as well as a skyline view. The fit of the metal components was assessed, and a deviation of ≥1 mm overhang/underhang was considered as abnormal.

### 2.5. Statistical Analysis

Demographic information including age, BMI, and gender is presented as descriptive statistics, i.e., averages, proportions, minimum and maximum values. Outcome measures such as ROM, KSS, WOMAC, and SF-12 scores are presented as descriptive statistics using averages, ranges, and standard deviations (SD). To determine the significance of changes in outcome measures between follow-up time points, a two -ailed Student’s t-test assuming unequal variances was performed, since the study reports on longitudinal data within the same cohort comparing pre- to respective postoperative data. A *p*-value of <0.05 was considered to indicate statistical significance. All statistical analyses were conducted using the build-in functions in Microsoft Excel (Microsoft Inc., Redmond, WA, USA).

## 3. Results

### 3.1. Study Population and Intraoperative Parameters

Patients’ demographics are shown in Table 1. Of the 73 patients included, 41 required right-knee implants, and 32 left implants; the average age of the patient population was 64 years (range: 40–81), the average BMI was 31 (range: 23–42), while 28 female and 45 male patients were recruited (Table 1). In 37 (51%) cases, surgery was performed under general anesthesia, and in 36 cases (49%), spinal anesthesia was administered. The average time of surgery was 93 min (range 66 to 142 min, SD 16.6).

### 3.2. Complications and Survial

By the time of the last follow-up, there had been one revision at 8 months postoperatively due to septic loosening of the implant, resulting in an implant survival rate of 98.6% at the time of the final follow-up. One patient required further surgical intervention due to progression of retro-patellar osteoarthritis, which included revision of the tibial plateau inserts and implantation of a patellar button. There were three manipulations under anesthesia (MUA) for reduced range of motion due to arthrofibrosis at 3, 4, and 9 months after surgery.

### 3.3. Outcome Parameters

Preoperatively, patients showed an average ROM of 106° (range, 70° to 125°, SD 14.4) for their knee to be treated (Figure 3). At 3 months post-OP, a similar ROM was observed, with an average of 106° (range, 75° to 125°, SD 12.3). From 3 to 6 months, a significant increase in ROM was observed, with an average of 112° (range, 85° to 125°, SD 11; *p* = 0.007) at 6 months post-OP. This significant increase in ROM continued during the following 6 months, with an average of 119° (range, 100° to 125°, SD 6.1; *p* = 0.003) observed 1 year post-OP. The range of motion after CCR-TKA reached a plateau at approximately 2 years, with averages of 122° (range, 105° to 125°, SD 5.7), 122° (range, 100° to 125°, SD 6.1), and 122° (range 115° to 125°, SD 4.7) at 2, 3, and 5 years post-OP, respectively (Figure 3).

When evaluating the results for the KSS-Function- and KSS-Knee-Score, a statistically significant increase was observed in both scores from pre-OP (Function: 53, Knee: 41) to 3 months (Function 74, Knee: 79; *p* < 0.001) and further to 6 months (Function: 84, Knee: 90; *p* < 0.001) post-OP (Figure 4). The score results reached a plateau 1 year after surgery (Function: 89, Knee: 92; *p* < 0.001) and showed no further significant changes during the follow-up period (Figure 4).

Similar results were observed in the analysis of the overall WOMAC-Score results. A significant decrease was observed from an average preoperative WOMAC-Score of 49 (range, 100 to 6, SD 20.9) to an average of 21 (range 80 to 0, SD 17; *p* < 0.001) at 3 months and an average of 13 (range, 80 to 0, SD 16; *p* = 0.002) at 6 months post-OP. The WOMAC-Score results at 6 months were not found significantly different from those at up to 5 years postoperatively, with an average of 9 (range, 56 to 0, SD 15.2) at 1 year, an average of 8 (range 36 to 0, SD 10) at 2 years, an average of 20 (range 72 to 0, SD 14) at 3 years, and an average of 17 (range 34 to 0, SD 8) at 5 years post-OP (Figure 5). The respective subscores WOMAC Pain, WOMAC Stiffness, and WOMAC Function followed this pattern, without major differences during the 5-year time course (Figure 5).

Lastly, in both, the SF-12-Physical- and the SF-12-Mental-Score showed an increase from the averages of 28 (range, 14 to 50, SD 7.7) and 49.9 (range, 29 to 69) preoperatively to averages of 39.8 (range, 22 to 54, SD 8.6) and 57.6 (range, 35 to 66, SD 7.1) at 3 months, averages of 45.4 (range 20 to 57, SD 8.6) and 57.9 (range, 29 to 64, SD 6.3) at 6 months, averages of 48.7 (range, 20 to 57, SD 9.5) and 56.5 (range, 43 to 65, SD 4.5) at 1 year, averages of 45.3 (range, 15 to 56, SD 11) and 52.5 (range, 34 to 60, SD 6.5) at 2 years, averages of 44.2 (range, 15 to 57, SD 11) and 53.8 (range, 35 to 61, SD 5.9) at 3 years, and averages of 50 (range, 32 to 57, SD 9.1) and 52.5 (range, 32 to 61, SD 9.5) at 5 years post-OP, respectively (Figure 6). However, only differences in the physical part of the SF-12 were found statistically significant (*p* < 0.001).

All knees treated were routinely evaluated radiographically pre- and postoperatively and showed and ideal implant fit without any over- or underhang of the components.

## 4. Discussion

TKA is performed increasingly more often in relation to the high demands of physical activity and quality of life even in old ages. However, around 20–30% of TKA patients remain unsatisfied with their surgical procedure, and factors such as implant malalignment and component oversizing have shown to be among the most common reasons for postoperative complaints [2,4]. Moreover, the clinical outcomes after TKA surgery in the young patient population are worse than previously thought, with high expectations of the surgical procedure being evident [20,21]. This discrepancy between unsatisfactory results and rising expectations for the TKA procedure is driving implant manufacturers and surgeons alike to improve the surgical techniques as well as the instruments and implants used [5,7]. Recent studies have highlighted the variability of knee joint geometry among individuals [22,23]. The idea of implant customization is thought to address this variability. In this context, the C-TKA used in this study aims to achieve an individual implant fit, to recreate patients’ individual joint geometry and kinematics, and therefore to improve the postoperative clinical and functional outcomes [8]. This is facilitated by manufacturing the femoral implant according to the individual J-curve anatomy of the patient derived from preoperative computed-tomography (CT) scans of the knee. The accuracy of the femoral component positioning in this CCR-TKA system can be further confirmed by the agreement of the measured thickness of the posterior femoral condyle resection with the preoperative iView^®^ plan (Figure 2). The tibial component is designed to restore the patient-specific slope and the asymmetric tibial plateau, while the ligament balance may be fine-tuned using different insert heights medially and laterally [8]. According to initial biomechanical studies, this individualized design approach may lead to the restoration of more physiological knee kinematics compared to OTS implants in vitro [24,25] and in vivo [26].

This implant system has been shown to facilitate precise implant positioning and correction of the mechanical axis in radiographic studies [9,27]. The data presented in these studies are consistent with the results of an intraoperative observational study that compared the rotational alignment of the tibial component according to the method of Cobb et al. using this CCR-TKA implant system with that achieved with various OTS-TKA systems and demonstrated improved tibial component rotation with preserved cortical bone coverage for the CCR-TKA [28]. As both implant fit and rotational alignment have been shown to correlate with clinical outcomes and satisfaction [29], these previous radiological findings are in agreement with the results of our analysis that show very good patient-reported outcomes and high satisfaction at mid-term follow-up.

To date, only limited clinical data are available on this CCR-TKA system. In an intra-hospital analysis, reduced blood loss and length of stay (LOS) in the hospital has been observed for CCR-TKA compared with OTS implants at comparable tourniquet and surgical times in a cohort of 621 patients [30]. A different study compared selected hospital outcomes in a consecutive series of patients undergoing TKA using either a customized implant or an OTS implant design. The authors observed significantly lower transfusion rates and fewer adverse events at discharge and after 90 days [24]. Although hospital metrics after CCR-TKA were not assessed in this single-arm study, promising observations regarding perioperative data using CCR-TKA made in these previous studies may have been associated with high patient-reported outcomes that were systematically obtained up to 5 years post-OP in this study cohort.

This study has several limitations, and the results should be interpreted accordingly. First, there was no control group for comparison, and therefore no comparison could be made between outcomes of patients with CCR-TKA and those of patients with OTS-TKA or with TKA implanted utilizing different alignment philosophies such as mechanical or kinematic alignment. Second, there was no blinding at any stage of the procedure, and therefore the high patient-reported outcomes may have been influenced by the patient’s awareness of having received a custom-made implant. Third, the OA knees included were not phenotyped according to alignment principles, which are currently under discussion. We believe that varus and valgus deformities of varying severity should be considered distinct entities of the disease, as proposed by the group of Hirschmann et al. [31]. In addition, a larger number of patients and longer study periods are needed to delineate and correlate results with phenotypes and surgical approaches. Lastly, because each implant is unique, general statements about this implant system are difficult to make, although the principles of mechanical axis reconstruction, joint line restoration, and J-curve anatomy are valid and incorporated into each CCR-TKA implant.

However, taking these limitations into account, high clinical outcomes were observed with CCR-TKA that can be compared with those of other OTS-TKA systems at mid-term follow-up. Regarding postoperative ROM, CCR-TKA showed similar or even better results compared to a previous study that assessed ROM for the Attune PS und Press-Fit-Condylar (PFC) Sigma PS TKA systems (DePuy/Synthes, New Brunswick, NJ, USA; 5° vs. 16°) at 2 years of follow-up [32], which are thought to have even higher ROM compared to the respective CR versions. Similarly, patients treated with CCR-TKA showed constantly high ROM (122° at 3–5 years FU; Figure 3), which is in contrast to results observed with the Scorpio CR TKA system, which showed a decrease in ROM 2 years postoperatively [33]. In a different study on the Genesis II TKA system (Fa. Smith & Nephew, London, UK), the authors reported worse ROM at 5 years postoperatively than in this study with the CCR-TKA (114° vs. 122°) [34].

The KSS scoring results observed with the CCR-TKA were somewhat comparable to those with the Attune PS und PFC Sigma PS TKA systems (DePuy Synthes) at 2 and 3 years [32], the Genesis II TKA system [34], and the Attune system using a rotating platform (DePuySynthes) [35] at 5 years postoperatively. Palmer et al. reported almost similar “Knee-subscores” with the Triathlon CR (Stryker Orthopaedics, Kalamazoo, MI, USA), but inferior results compared to the CCR-TKA in the “Function-subscore” (72 vs. 83 points) 2 years postoperatively [36].

In the studies by Harato et al. and Powell et al., WOMAC scores for the PFC Sigma CR or the Genesis II TKA systems at 5 years of follow-up were similar to those in this study [34,35]. In contrast Chaudhary et al. observed worse subscores for “Pain” (14.9 vs. 13.7 points) and “Function” (23.5 vs. 14.9 points) at 2 years postoperatively with the Scorpio CR TKA system compared to the results with the CCR-TKA [33].

Regarding the SF-12 scores, the Genesis II TKA system (Smith & Nephew Inc.) showed worse results in the PCS subscore (42.5 vs. 50.0 points), while the MCS subscore was the same compared to the results observed with the CCR-TKA at the same postoperative timepoint [34]. The PFC Sigma (DePuy Synthes) in comparison showed worse results for both PCS and MCS subscores [35], and the Triathlon CR (Stryker Inc.) showed worse score results for the PCS, while the MCS was at the same level [37] compared with the CCR-TKA at 5 years of follow-up.

However, none of these studies directly compared CCR-TKA with OTS-TKA, and therefore the power of all these comparisons is limited, because different study populations and observers were involved. Furthermore, it may be doubted that differences in postoperative outcomes between TKA knee designs can be elucidated by the use of quality-of-life scores such as the SF-12 or the WOMAC, as they are influenced by a variety of other factors besides knee kinematics and perception of the artificial knee. Perhaps, more refined patient-reported outcome measures such as the “Forgotten Joint Score” may be able to elaborate such differences, especially when they are used in a comparative and blinded fashion over longer periods of time [38]. Further, more refined analytical tools such as regression analysis could be applied in a comparative investigational setup, allowing for more detailed information in which variables matter most in the comparison of different TKA designs and alignment principles and of the extent and level of confidence.

Nonetheless, this study presents promising mid-term clinical outcome data using this patient-specific implant system. Our observations are also supported by the Orthopaedic Data Evaluation Panel in the United Kingdom (ODEP) (http://www.odep.org.uk, accessed on 25 January 2022), who awarded the Conformis iTotal^®^ CR knee replacement system a “5A” rating, based on strong evidence of implant performance over 5 years, including low revision rates as indicated in the UK’s National Joint Registry (NJR).

## 5. Conclusions

In summary, this study demonstrates that good clinical outcomes may be achieved with this CT-based fixed-bearing CCR-TKA system, which are consistent with existing radiological, preclinical, and early clinical data. Further follow-up studies, including long-term comparative studies of clinical outcomes parameters, are needed to clarify whether this system is a valuable alternative treatment modality for patients with tricompartmental knee OA.

## Figures and Tables

**Figure 1 jpm-12-00194-f001:**
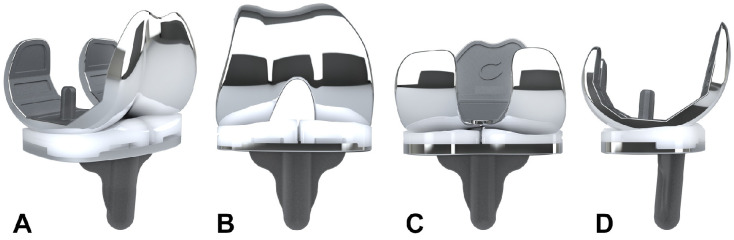
Representative images of the patient-specific, cruciate-retaining iTotal CR G2 total knee replacement. (**A**) Antero-medial, (**B**) anterior, (**C**) posterior, and (**D**) lateral view of the device. Please note that the medial and lateral polyethylene inserts have different heights that correspond to the condylar offset of the femur.

**Figure 2 jpm-12-00194-f002:**
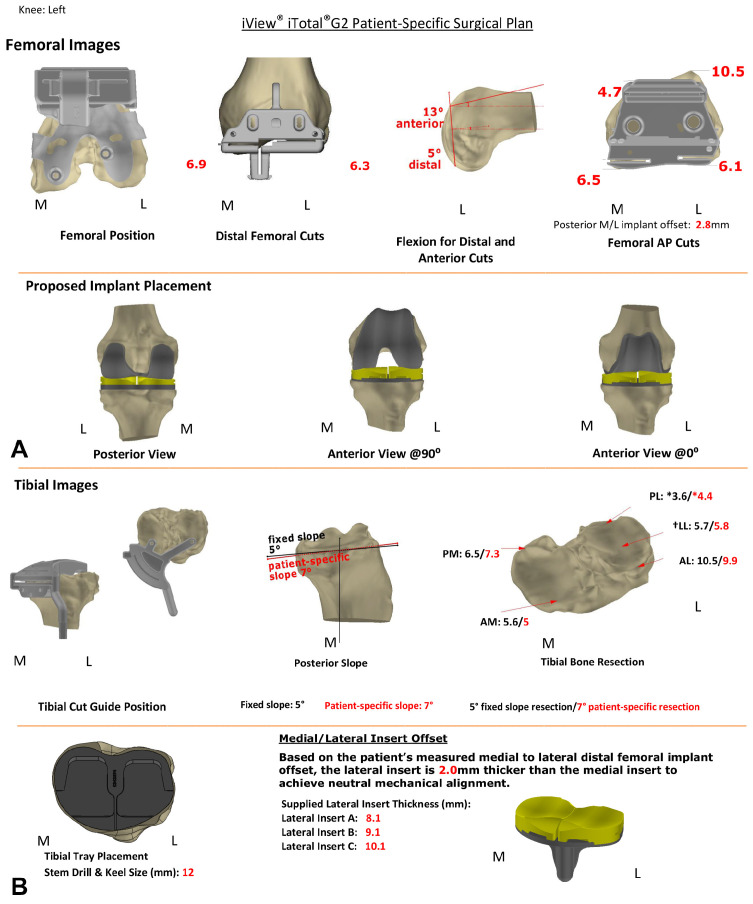
Representative patient-specific surgical plan (iView^®^). Upper images (**A**) show the positioning of the respective drill guide, cutting jigs, and implant for femoral preparation, and the projected thickness of the cut bone is given in red for self-control. Lower images (**B**) show the positioning of the respective cutting jig and implant for tibial preparation, and the projected thickness of the bone cut is given for self-control. Please note that a fixed cut of 5° slope (black numbers) or a patient-specific slope cut guide (here 7°; red numbers) can be chosen, and asterisks (*) indicate a point 5 mm from edge.

**Figure 3 jpm-12-00194-f003:**
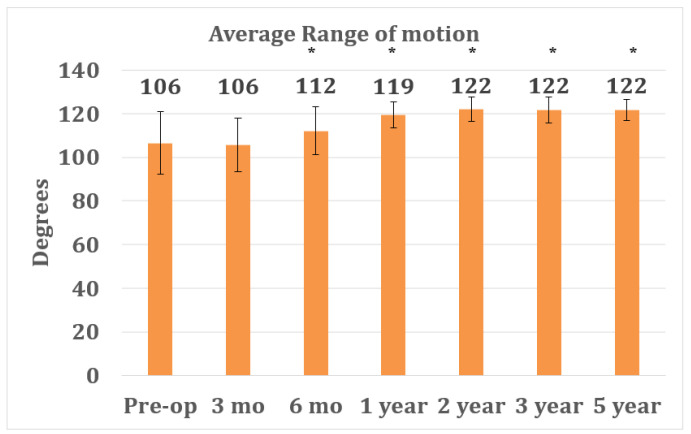
Average range of motion at pre-operative and follow-up time points after CCR-TKA surgery. The graph reports averages +/− SD. Asterisks (*) indicate statistical significance compared with the pre-op control, as determined by Student’s *t*-test (*p* < 0.05).

**Figure 4 jpm-12-00194-f004:**
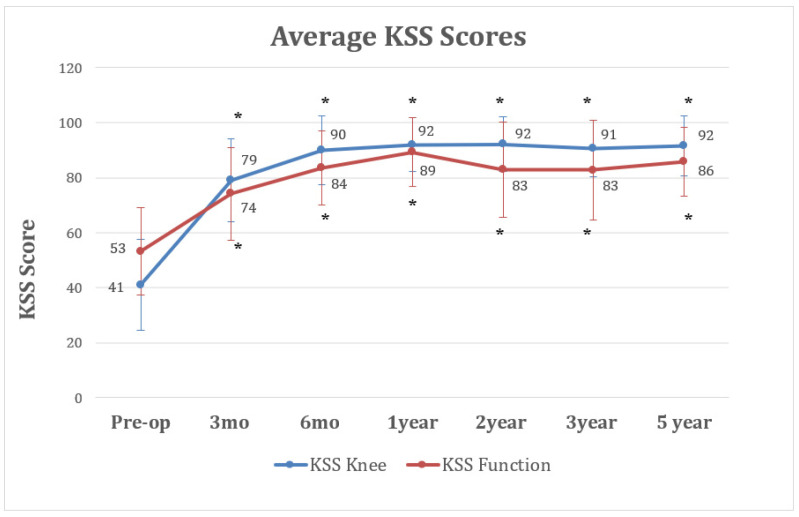
Average Knee Society Scores (KSS) for patients enrolled in the study, including the knee (blue) and function (red) subscores. The graphs reports averages +/− SD. Asterisks (*) indicate statistical significance compared with the pre-op control, as determined by Student’s *t*-test (*p* < 0.05).

**Figure 5 jpm-12-00194-f005:**
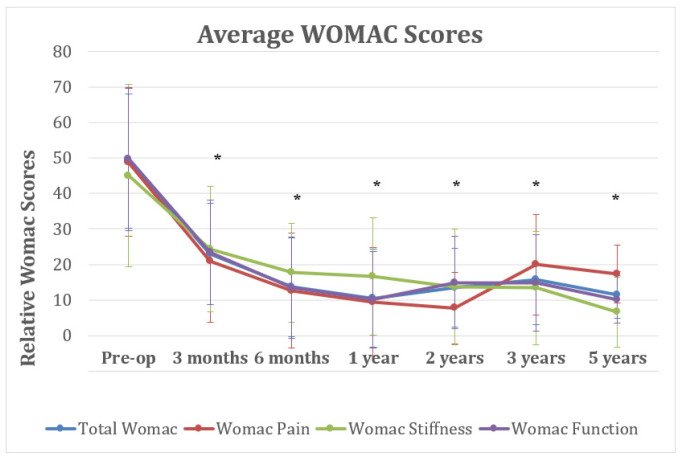
Average Western Ontario and Mc Master Index (WOMAC) for patients enrolled in the study, including the total score (blue), as well as the pain (red), stiffness (green), and function (blue) subscores over time. The graphs reports averages +/− SD. Asterisks (*) indicate statistical significance for all WOMAC scores (total WOMAC, WOMAC pain, WOMAC Stiffness, WOMAC Function) at the respective timepoint compared with the pre-op control scores, as determined by Student’s *t*-test (*p* < 0.05).

**Figure 6 jpm-12-00194-f006:**
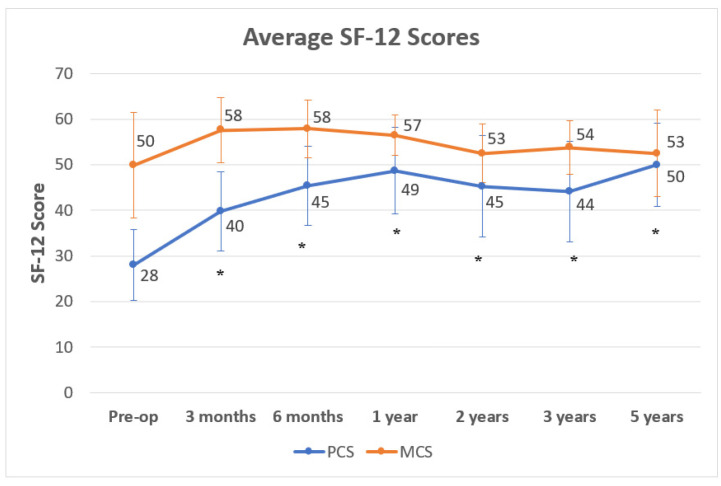
Average Short Form (SF)-12 scores from the pre-operative time point up to the 5-year post-operative time point for patients enrolled in the study, including physical (PCS, blue) and mental (MCS, orange) subscores. The graph reports averages +/− SD. Asterisks (*) indicate statistical significance in comparison with the pre-op control, as determined by Student’s *t*-test (*p* < 0.05).

**Table 1 jpm-12-00194-t001:** Demographic profile of patients enrolled in the study.

Metric		Min	Max
Knees (N)	73	-	-
Patient Gender (% Female)	38%	-	-
Mean Age at Surgery (years)	64	40	81
Mean Body Mass Index (BMI)	31	23	42

## Data Availability

The datasets used and/or analyzed during the current study are available from the corresponding author on reasonable request.

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
