# Peer review of "The Impact of Total Knee Replacement with a Customized Cruciate-Retaining Implant Design on Patient-Reported and Functional Outcomes"

_jpm, 2022, doi:10.3390/jpm12020194_

Round 1

Reviewer 1 Report

The title is partly inappropriate, the functional impact after KTA treatment seems almost obvious, but "excellent" is misleading and unscientific.. I suggest "the impact of .... on functional outcome"

 I strongly suggest evaluating the CROSS guidelines https://doi.org/10.1007/s11606-021-06737-1

 Abstract:

  In the methods it is more appropriate not to include the results of the selection and subsequently the eligibility .. a) design b) participants c) outcome ..

 Level 1 can mean several concepts. In other countries .. contexts. The approval and consent is not necessary in the abstract. If possible put the registration codes at the bottom of the abstract.

  39-42 results, not methods.

 The intervention performed in the study is missing

 The first sentence of the conclusions is strictly methodological in nature

 Introduction

 Increase the number of references

 Methods

 96-98 are once again results not methods

 Describe the interventions, scale score (e.g., number of sections, number of questions, number and names of instruments used).

 Describe the study population (i.e., background, locations, eligibility criteria for participant inclusion in survey, exclusion criteria). Moreover, Describe the sampling techniques used (e.g., single stage or multistage sampling, simple random sampling, stratified sampling, cluster sampling, convenience sampling). Specify the locations of sample participants whenever clustered sampling was applied.

But Most of all: Describe statistical methods and analytical approach. Normality test? Why T-T test? Paired? Like a pre-post comparison? I believe that relying on the Excel suite alone seems insufficient for the power of calculation In this regard the graphs are of little scientific impact there is no SD or SE ..

beyond the discussion, I can recommend the evaluation of a regression analysis of the single items of the scores for example the PCS - SF12 .. what changes, when it changes.. Honestly the legitimacy of the fact the most common scores improve in patients after KTA's intervention does not provide the current literature with new perspectives .. but a good sample with deeper investigations on the results could make us analyze new ideas.

for this reason I recommend a deep major review

Author Response

Dear Reviewers,

Thank you for the very helpful comments. We have extensively revised our manuscript according to these comments, including the title, presentation of the data and research design, introduction, description of the methods, presentation of the results as well as the conclusions. The specific responses to the comments are as follows:

Reviewer #1

Comment #1: The title is partly inappropriate, the functional impact after KTA treatment seems almost obvious, but "excellent" is misleading and unscientific. I suggest "the impact of .... on functional outcome"

 We agree and changed the Title accordingly:

„The impact of total knee replacement with a customised cruciate-retaining implant design on patient reported and functional outcome”

Comment #2: I strongly suggest evaluating the CROSS guidelines https://doi.org /10.1007/s11606-021-06737-1.

This is a very valuable article and checklist. Thank you for sharing. We tried to address as many points as possible throughout the manuscript. The specifics are mentioned in the comments below.

Comment #3: Abstract: In the methods it is more appropriate not to include the results of the selection and subsequently the eligibility. a) design b) participants c) outcome. Level 1 can mean several concepts. In other countries . contexts. The approval and consent is not necessary in the abstract. If possible put the registration codes at the bottom of the abstract. 39-42 results, not methods. The intervention performed in the study is missing. The first sentence of the conclusions is strictly methodological in nature

All these points have been addressed and the Abstract was now modified accordingly (lines 36-60). The results of the section Methods has been moved to the Results section. “Level 1” was deleted to avoid confusion. Approval and consent have also been deleted. Lines “39-42” were moved to the results. The intervention is now included: “At our arthroplasty center we prospectively analysed a cohort of patients with tricompartmental gonarthosis to be treated with total knee replacement (TKA) using a customised (C-) cruciate retaining (CR) implant design” (lines 41-43). The first sentence of the conclusions was deleted.

Comment #4:  Introduction:  Increase the number of references

We now increased the number of references in the introduction from 7 to 14. The specific new references can be found in the section References.

Comment #5: Methods:

  1. Lines 96-98 are once again results not methods

Former lines 96-98 as well as the Table 1 were now moved to the Results section (lines 463-466)

  1. Describe the interventions, scale score (e.g., number of sections, number of questions, number and names of instruments used).

The intervention by means of surgical intervention is described in the Methods section lines 324-393. Within this text we now included the number of surgical steps and customised iJig instruments with their names.  

“…the surgical procedure includes 6 different steps, that are facilitated by the use of the provided patient-specific bone resection jigs and iView® protocol, allowing confirmation of all performed bone cuts against the surgical plan for self-control. Specifically, the instrumentation kit comprises 6 different femoral (F1-6), and 5 separate tibial (T1-5) jigs for cutting and drilling. The surgical steps are the distal femoral resection (required jigs: F1-3), the proximal tibial resection (jig: T1), balancing of extension and flexion gap (jigs: T2, T3), femoral preparation (jigs: F4, F5), trialing (jigs: F6, T4), and final tibial preparation (jig: T5), before the final implantation of the components is performed. Notably, the surgeon has two options for the tibial cuts with two different T1 instruments facilitating the tibial cut either with a patient specific slope (shown in red) or a fixed slope of 5° (shown in black) as shown on the tibial images of the iView® (Figure 2A). Additionally, traling can be performed with three individually designed T4 jigs with 1 mm incremental heights. Thereby, the C-TKA system allows for kinematic testing using anatomic trial components, where the most appropriate tibial insert heights for the medial (6 mm, 7 mm, or 8 mm) and lateral (A, B, C with custom thicknesses; see iView® bottom row (Fig. 2)) knee joint space may be identified before the final components are implanted.”

The interventions by means of scores and parameters tested over time are described in the Methods section lines 343-414. Within this text we now included more specifics of the test instruments (scores) used.  

“The Knee Society Scoring System (KSS) by Insall et al. was assessed that consists of 2 separate subscales: (1) A "Knee" score (100 points total) which considers pain (50 points), stability (25 points) and range of motion (25 points) with deductions for flexion contractures, extension lag and malalignment. (2) A "Function" score (100 points total) that utilizes walking distance (50 points) and stair climbing (50 points) with deduction for the use of a walking aid [15,16].

              The Western Ontario and McMaster Universities Osteoarthritis Index (WOMAC) was used as a patient reported outcome measure for knee osteoarthritis, that included the subscales "Pain" (5 items; 50 points), "Stiffness" (2 items; 20 points) and "Function" (17 items; 170 points), with a range from 0 (= no pain/stiffness/problems) to 10 (= extreme pain/stiffness/impossible to do) points for each item [17]. The relative WOMAC scores, i.e. for the total WOMAC score as well as for the subscales, are then calculated from the point values by multiplication x 100 and divided by the maximum score value [17,18].

              We also assessed the Short Form (SF) 12 Health Survey, which is a 12-item, patient-reported survey of patient health, that evaluates eight dimensions of health status [19]. Scores 0–100 are given for each subscale, and higher scores represent better health. Norm-based scoring of each 0–100 scale is then carried out by standardisation of each subscale reltive to a Z-Score that is 50 on average in the popuation with a standard deviation of 10 [19]. Finally, two aggregate summary measures can be derived, a physical (PCS) and a mental (MCS) health subscore to determine overall mental and physical well-being [19].”

  1. c) Describe the study population (i.e., background, locations, eligibility criteria for participant inclusion in survey, exclusion criteria). Moreover, describe the sampling techniques used (e.g., single stage or multistage sampling, simple random sampling, stratified sampling, cluster sampling, convenience sampling). Specify the locations of sample participants whenever clustered sampling was applied.

The study population is described in the section Methods / 2.1 Patients (lines 195-207) with all inclusion and exclusion criteria. Sampling was performed on consecutive patients that met the inclusion criteria and willing to participate in the study. Nowadays in Germany patients are usually not performing the postoperative controls in the hospital but in private practice or outpatient clinics at the referring physician. However, this limited cohort agreed to participate in this follow-up study at our surgical unit. We now clarified the sampling within this section (lines 207-209):

“Patient enrollment and sampling was conducted to include subsequent cases willing to participate and to meet the inclusion criteria after a learning curve with this implant system of 6 months.”

  1. d) Describe statistical methods and analytical approach. Normality test? Why t-test? Paired? Like a pre-post comparison? I believe that relying on the Excel suite alone seems insufficient for the power of calculation. In this regard the graphs are of little scientific impact there is no SD or SE.

The statistical analysis is now described in the section Methods in detail as requested (lines 453 - 460). 

Demographic information including age, BMI and gender is presented as descriptive statistics, i.e. averages, proportions, minimum and maximum values. Outcome measures such as ROM, KSS, WOMAC, or SF-12 scores are presented as descriptive statistics using averages, ranges and standard deviations. To determine significance of changes in outcome measures between follow-up time-points, a two tailed Student’s t-test assuming unequal variances was performed, since the study reports on longitudinal data within the same cohort comparing pre- to respective postoperative data. A p-value of <0.05 was considered to indicate statistical significance. All statistical analysis was conducted using pre-built functions in Microsoft Excel (Microsoft Inc, Redmond-WA, USA).

  1. e) Beyond the discussion, I can recommend the evaluation of a regression analysis of the single items of the scores for example the PCS - SF12. what changes, when it changes. Honestly the legitimacy of the fact the most common scores improve in patients after TKA's intervention does not provide the current literature with new perspectives, but a good sample with deeper investigations on the results could make us analyze new ideas.

We entirely agree. However, as no control group(s) of any alternative treatment modality was present within this study and "only" a prospective longitudinal study of the respective outcome variables following C-TKA was performed, a regression analysis was not included here. However, we now added this point to the Discussion (lines 933 - 935):

"Further, more refined analytical tools such as regression analysis could be applied in a comparative investigational setup, allowing for more detailed information on which variables matter most in C-TKA compared with OTS-TKA, and to what extent, and how confident we can be about these variables."   

Reviewer 2 Report

Dear Authors,

this is more or less an observational study.

I miss a clear study design with study questions that are going to be answered or one or more hypotheses, that connected to the personalized design of this knee prosthesis. Just collecting data and concluding that this kind of knee prosthesis design is superior is not enough.

Furthermore, the manuscript has a number of orthographic errors that need to be revised.

Author Response

Dear Reviewers,

Thank you for the very helpful comments. We have extensively revised our manuscript according to these comments, including the title, presentation of the data and research design, introduction, description of the methods, presentation of the results as well as the conclusions. The changes in the revised manuscript are in RED COLOR. The specific responses to the comments are as follows:

Reviewer 2:

Comment #1: This is more or less an observational study. I miss a clear study design with study questions that are going to be answered or one or more hypotheses, that connected to the personalized design of this knee prosthesis. Just collecting data and concluding that this kind of knee prosthesis design is superior is not enough.

We agree that this is an observational study in which we focus on the development of the clinical parameters over time after C-TKA surgery - not more and not less. As no mid-term clinical data of such personalised C-TKA systems is present thus far in the literature, we still think our data is of strong interest to the readers of the PM journal. By nature, the scientific value of such case studies is limited (EBM level 3b), as any control group is missing for comparison. We highlighted these limitations extensively in the Discussion (lines 926-933).

Comment #2: Furthermore, the manuscript has a number of orthographic errors that need to be revised.

We corrected all orthographic and grammatical errors throughout the manuscript and implemented all suggestions which were given by the reviewer within the text. Responses to specific comments are as follows:

Line 276 these are nor "cruciate retaining" systems; are they comparable? If yes, why?

Posteror-stabelised (PS) TKA systems are thought to have a slightly higher ROM compared to their CR versions and therefore the comparison to these systems is of interest. We added this to the text of the Discussion (line 792):  

“, which are thought to have even higher ROM compared to the respective CR versions”

Line 309: you did not present any radiographic data; are such data collected and published in another Journal? Are any radiological parameters measured?

Yes, in depth radiological data have been published elsewhere (see reference [9]). That is why we here only included the fact in that any over- or underhang (<>1mm) of the components was absent in the Results section (lines 555-556).

Having addressed the reviewer comments, we would appreciate your reconsideration of our paper in the light of these changes.

Thank you for consideration.

Round 2

Reviewer 1 Report

Thanks for the effort after a first brutal review. I can only suggest better describing the abbreviation as customized cruciate retaining total knee arthroplasty as CCR-TKA and not as C-TKA as it could be confused with the more common term of Conventional TKA...

Best Regards

Author Response

Dear reviewer,

thank you again for your comments. We now changed the abbreviation "C-TKA" to "CCR-TKA" throughout the manuscript as suggested. 

Thank you again for considering our manuscript.

Reviewer 2 Report

  • Still a number of orthographic errors
  • Still not a well written paper that reports about an investigation with clear questions, clear answers, clear conclusion

Author Response

Dear reviewer,

Thank you so much for your suggestions, which now have all been implemented within the new manuscript.

Thank you so much for your reconsideration.

Round 3

Reviewer 2 Report

Dear Authors,

there is a wide interest in papers like yours. 

So, I do support publication of your manuscript .

See my final comments.
